# MLP-Attention: Improving Transformer Architecture with MLP Attention Weights

**Alireza Morsali** *
Independent Research Scientist
Vancouver, BC, Canada
`alireza.morsali@mail.mcgill.ca`

**Moein Heidari & Samin Heydarian1 & Tohid Abedini**
Iran University of Science and Technology
Tehran, Iran
`moeinheidari7829@gmail.com`
`{samin_heydarian,t_abedini}@comp.iust.ac.ir`

## Abstract

The Transformer architecture has revolutionized natural language processing (NLP) and has achieved state-of-the-art results in various tasks. The attention mechanism is one of the key components of the Transformer architecture, which allows the model to focus on relevant parts of the input. In the standard Transformer, the attention weights are computed by the dot product of query and key vectors followed by a softmax function. However, in this paper, we propose to replace the dot product of query and key vectors with a multi-layer perceptron (MLP) to compute attention weights directly from the embeddings. The proposed modification is simple and can be easily implemented in existing Transformer-based models to improve their performance as shown in this paper for an NLP task. We provide the implementation code at https://github.com/AlirezaMorsali/MLP-Attention for reproducibility and ease of adoption.

## 1 Introduction and Related Work

The Transformer model Vaswani et al. (2017) has garnered widespread adoption in diverse fields, particularly in natural language processing, and has emerged as the preferred architecture for pre-trained models Qiu et al. (2020). Its utility has also been recognized in computer vision Parmar et al. (2018), audio processing Dong et al. (2018), chemistry Schwaller et al. (2019), and life sciences Rives et al. (2021). Despite its broad applicability, the Transformer faces certain challenges such as inefficiency in processing long sequences, difficulty in training on small-scale data, and the need for adaptation to specific downstream tasks Liu et al. (2018). Several modifications have been proposed to address these challenges, with a focus on enhancing model efficiency, generalization, and adaptation through alterations to the architecture, pre-training methods, and downstream applications Child et al. (2019); Katharopoulos et al. (2020).

Shen & Huang (2016) proposed a novel attention-based convolutional neural network architecture for relation classification. Their model utilized word embedding, part-of-speech tag embedding, and position embedding information, along with a word level attention mechanism to better determine which parts of the sentence are most influential with respect to the two entities of interest. Guo et al. (2022) introduced an attention mechanism called external attention. Their method is based on two external, small, learnable, shared memories, which can be implemented easily by simply using two cascaded linear layers and two normalization layers. The authors incorporated the multi-head mechanism into external attention to provide an all-MLP architecture, called external attention MLP (EAMLP), for image classification. Liu et al. (2021) proposed a simple network architecture, gMLP, based on MLPs with gating, and showed that it can perform as well as Transformers in key language and vision applications. The authors demonstrated that self-attention is not critical for Vision Transformers, as gMLP can achieve the same accuracy.

These papers explore various aspects of attention mechanisms, including different types of attention mechanisms, their efficiency, and the applicability of MLP-based models as alternatives to attention-based models. However, none of these papers investigate the idea proposed in our paper, which suggests using an MLP-based attention mechanism in transformer architecture.

---

*https://alirezamorsali.github.io/.

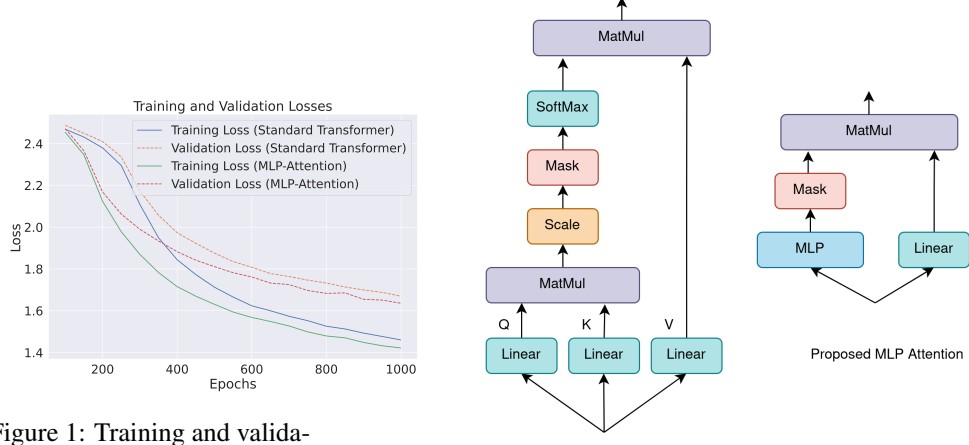

Figure 1: Training and validation loss

Figure 2: Block diagram of standard dot product attention and proposed MLP attention.

## 2 METHODOLOGY

We propose an MLP-Attention model that employs a multi-layer perceptron (MLP) to compute attention weights for sequences of word embeddings, optionally with positional encodings. The MLP is designed to perform multiple layers of nonlinear transformations on the input, and its output serves as the attention weights. The MLP's architecture can be customized to meet the task's specific requirements.

To simplify the transformer architecture and improve accuracy during both training and inference, we eliminate the query and key vectors and directly calculate the attention weights. Figure 2 depicts the block diagram of the proposed MLP attention model and the standard dot product attention. Additional details about the MLP-Attention architecture are provided in Appendix A.

Our proposed MLP-Attention model provides a flexible and efficient alternative to the traditional transformer architecture, allowing for customization of the MLP architecture to meet specific task requirements while also simplifying the overall model.

## 3 EXPERIMENTS

We conducted an experimental evaluation of our proposed MLP-Attention model, which is designed for character prediction tasks using the Tiny Shakespeare dataset (https://github.com/karpathy/char-rnn/). Our study compared the performance of our model against the vanilla Transformer model, and we analyzed the training and validation loss of both models, as depicted in Figure 1. The code used for implementation and evaluation can be found in the GitHub repository at https://github.com/AlirezaMorsali/MLP-Attention. Additional details about the experiments are provided in Appendix. B.

Our results demonstrate that the MLP-Attention model outperforms the Transformer model in terms of accuracy, with only 10% more parameters than the Transformer model. Specifically, our MLP-Attention model has 6 million parameters, while the Transformer model has 5.5 million parameters.

Overall, our experiments provide compelling evidence for the effectiveness of the MLP-Attention model in natural language processing (NLP) tasks, particularly character prediction. These results demonstrate the potential of our proposed model to enhance the accuracy and efficiency of existing NLP models, which can benefit a wide range of practical applications.

## 4 CONCLUSION

In this paper, we propose a modification to the attention mechanism of the Transformer architecture by replacing the dot product of query and key vectors with an MLP. We show that our proposed MLP-Attention model outperforms the standard Transformer and other attention mechanisms on NLP tasks.

URM STATEMENT

The authors acknowledge that at least one key author of this work meets the URM criteria of ICLR 2023 Tiny Papers Track.

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

## A  ARCHITECTURE DETAILS

In this section, we provide a detailed description of our method. The core idea of our proposed model is to replace the dot product of query and key vectors with a multi-layer perceptron (MLP) to compute attention weights directly from the embeddings. This modification simplifies the architecture and enhances the accuracy during both training and inference.

### A.1  COMPUTING ATTENTION WEIGHTS

To calculate attention weights using the MLP, we follow the steps outlined below:

Input Sequences: We start with input sequences of word embeddings, which can be optionally accompanied by positional encodings. Let $\mathbf{X} \in \mathbb{R}^{T \times d}$ denote the input sequence matrix, where $T$ is the sequence length and $d$ is the embedding dimension.

MLP Transformation: The sequences are fed into an MLP, which performs multiple layers of nonlinear transformations on the input. Specifically, we apply $L$ layers of transformations, denoted as $\text{MLP}_l$, where $l \in 1, 2, ..., L$. Each MLP layer applies a linear transformation followed by a non-linear activation function, such as ReLU or GELU. The output of the $l$-th layer is computed as follows:

$$\mathbf{H}_l = \text{MLP}l(\mathbf{H}_{l-1}) = \text{ReLU}(\mathbf{H}_{l-1}\mathbf{W}_l + \mathbf{b}_l), \tag{1}$$

where $\mathbf{H}_l$ is the input to the $l$-th layer and $\mathbf{H}_0 = \mathbf{X}$, $\mathbf{W}_l$ and $\mathbf{b}_l$ are the weight matrix and bias vector of the $l$-th layer, respectively.

Output as Attention Weights: The output of the last MLP layer serves as the attention weights, replacing the traditional dot product calculation, i.e., $\mathbf{Q}\mathbf{K}^T$. These attention weights determine the importance of each word in the sequence during the attention mechanism. Additionally, The attention weights can be optionally masked if required similar to the traditional dot product attention weights. Let $\mathbf{A} \in \mathbb{R}^{T \times T}$ denote the attention weight matrix, computed as:

$$\mathbf{A} = \text{softmax}(\mathbf{H}_L), \tag{2}$$

where $\text{softmax}(\cdot)$ applies the softmax function along the rows of $\mathbf{H}_L$.

### A.2  MLP ARCHITECTURE AND CUSTOMIZATION

The architecture of the MLP can be adjusted to suit the needs of the task at hand. By modifying the number of layers, hidden units, and activation functions, the MLP can be customized to optimize performance. However, we note that the specific details of the MLP architecture, such as dimensionality and activation functions, are task-dependent and should be chosen based on empirical experimentation.

## B  EXPERIMENT DETAILS

In this section, we present the details of experimental results comparing the performance of the MLP-Attention model with the vanilla Transformer model. The experiments were conducted on a language modeling task using the Tiny Shakespeare dataset.

- Model Architecture: Vanilla Transformer: The baseline model consisted of a Transformer architecture with 3 layers each with 3 heads, and an embedding size of 384. The model utilized self-attention and feed-forward layers.
- MLP-Attention: The proposed MLP-Attention model had the same architecture as the baseline Transformer but with MLP-Attention layers instead of self-attention layers. For this experiment, an MLP with one hidden layer of size 256 is used with ReLU activation function.
- Hyperparameters:
  - Batch Size: 64
  - Block Size (maximum context length): 256
  - Learning Rate: 0.0003

- **–** Dropout rate: 0.2
- **–** Epochs: 100
- **–** Evaluation Interval: 50

- Training and Evaluation: The training data was split into 90% for training and 10% for validation. The models were trained using the ADAM optimizer with the specified learning rate. The training loss and validation loss were computed during training to monitor the model's performance. The evaluation was performed every evaluation interval iterations, and the loss values were recorded for analysis.

