# OpenReview forum: "MLP-Attention: Improving Transformer Architecture with MLP Attention Weights"
_ICLR.cc/2023/TinyPapers — Submitted to Tiny Papers @ ICLR 2023_

### Official Review · Reviewer_7cth · 2023-03-25

**Confidence:** 1

**Summary Of Contributions:**

The author proposes a way to modify transformer architecture by replacing key and query in transformer architecture with MLP

**Rating:**

Great Start (GS): a submission which meets some of the reviewing criteria but has room for improvement

**Strengths And Weaknesses:**

Strength:
* Author proved through loss function that his theory performed well compared to transformer loss for train and test set\
 Weaknesses:
* Though the good performance of the model on train and test set the author has to perform analysis to different datasets to come to conclusion or use the same dataset that were used to train the transformer in order to come up sustainable theory.

**Suggested Changes:**

* More metrics need to be put in place other than loss function
* More description of the dataset used
* More description of the used transformer

---

> ### Author Response · Authors · 2023-05-31
> **The authors express gratitude for the reviewer's comments and recognition of the strength of their proposed modification to the transformer architecture. They agree with all of the reviewer's feedback. We tried our best to  incorporate all the suggestions into the revised version of the paper.**
>
> We would like to express our gratitude to the reviewer for their insightful comments and feedback. We are pleased to hear that our proposed modification to the transformer architecture has been recognized as a strength in the paper.
>
> We agree with the reviewer's suggestion to perform analysis on different datasets to further validate the performance of our model. Unfortunately, due to a lack of computation resources, our attempt to provide an evaluation of our findings on a larger dataset was not successful. Furthermore, we apologize for any confusion and want to clarify that the same dataset was used to train both the vanilla transformer and MLP attention for a fair comparison.
>
> In response to the reviewer's feedback on the dataset and training concerns, we recognize the significance of clarifying the dataset and training settings for the sake of reproducibility. In the updated version, we have included comprehensive information regarding the training settings, including details about the optimizer, learning rate, batch size, and the origin of the dataset. Furthermore, we have made our implementation of both methods publicly available on GitHub, which can be accessed at https://github.com/AlirezaMorsali/MLP-Attention.
>
> We appreciate the reviewer's recognition of our work as a great start and their valuable suggestions for improvement. We have incorporated these suggested changes in the revised version of the paper to enhance the clarity and completeness of our research.

---

### Meta-Review · Area_Chair_Mig4 · 2023-04-08

**Recommendation:** Invite to revise
**Confidence:** 3

**Metareview:**

This paper proposes a modification to the Transformer architecture by replacing the dot product of query and key vectors with an MLP to compute attention weights directly from the embeddings. The author demonstrates that this modification improves the performance of the model on NLP tasks.

However, the work has several weaknesses that need to be addressed. The lack of context and details make it difficult to understand the proposed method, and the author needs to revise the paper to improve its clarity and reproducibility. The author needs to provide a detailed description of the proposed method, including how to calculate the attention weights using an MLP, which dimensionality the MLP applied to the representation, how to causally mask the output of an MLP, and the computation/FLOPs used in the new MLP attention, etc. The author also needs to explain the dataset and training settings to ensure the reproducibility of the work.

Overall, this work has potential but requires significant revision to meet the Clarity, Correctness, and Reproducibility (CCR) criteria. Therefore, the paper is recommended as ``Invite to Revise''.

**Summary:**

This paper proposes a modification to the Transformer architecture by replacing the dot product of query and key vectors with an MLP to compute attention weights directly from the embeddings. The author demonstrates that this modification improves the performance of the model on NLP tasks.

**Reason For Not Giving A Higher Recommendation:**

This work has several weaknesses that need to be addressed.

1. The lack of context and details make it difficult to understand the proposed method, and the author needs to revise the paper to improve its clarity and reproducibility. The author needs to provide a detailed description of the proposed method, including how to calculate the attention weights using an MLP, which dimensionality the MLP applied to the representation, how to causally mask the output of an MLP, and the computation/FLOPs used in the new MLP attention, etc.
2. The author also needs to explain the dataset and training settings to ensure the reproducibility of the work.

**Reason For Not Giving A Lower Recommendation:**

N/A

---

> ### Author Response · Authors · 2023-05-31
> **The authors appreciate the reviewer's feedback on the lack of clarity and reproducibility in the proposed method. We tried our best to address these concerns in the revised version of the paper.**
>
> We would like to thank the reviewer for their valuable feedback and constructive comments on our paper. We acknowledge that there are areas where the clarity and reproducibility of our proposed method can be improved, and we addressed these concerns in the revised version of the paper.
>
> Regarding the lack of context and details, we apologize for any confusion caused. In the revised manuscript, we provided a more detailed description of the proposed method, including a step-by-step explanation of how attention weights are calculated using the MLP.
>
> To ensure reproducibility, we understand the importance of explaining the dataset and training settings. In the revised version, we also provided detailed information about the training settings, such as the optimizer, learning rate, batch size, and the source of the dataset. Moreover, we also publically published our implementation of both methods in the GitHub repository
> at https://github.com/AlirezaMorsali/MLP-Attention.
>
> We appreciate the reviewer's recognition of the potential of our work and their recommendation to revise the paper. We carefully addressed all the concerns raised and tried to provide a clearer and more reproducible description of our proposed method.

---

### Decision · Program_Chairs · 2023-04-08

Revision accepted; invite to archive